# Transcriptome Sequencing Analysis of circRNA in Skeletal Muscle between Fast- and Slow-Growing Chickens at Embryonic Stages

**DOI:** 10.3390/ani12223166

**Published:** 2022-11-16

**Authors:** Genxi Zhang, Jin Zhang, Pengfei Wu, Xuanze Ling, Qifan Wang, Kaizhi Zhou, Peifeng Li, Li Zhang, Hongxin Ye, Qi Zhang, Qingyu Wei, Tao Zhang, Xinglong Wang

**Affiliations:** 1College of Animal Science and Technology, Yangzhou University, Yangzhou 225009, China; 2College of Animal Science, Shanxi Agricultural University, Taiyuan 030032, China

**Keywords:** chicken, skeletal muscle, growth and development, circRNA

## Abstract

**Simple Summary:**

Broilers provide a large amount of protein for human every year, in which skeletal muscle contribute a major role. In recent years, circRNAs have been found to be involved in skeletal muscle development, and more significant circRNAs in the process need to be revealed. In the study, transcriptome sequencing technology was used to obtain differentially expressed circRNAs (DECs) between fast- and slow-growing chickens. A variety of bioinformatics analysis were then used to identify the key circRNAs including novel_circ_0004547, novel_circ_0003578, novel_circ_0010289 and novel_circ_0003082. In addition, some candidate circRNA-miRNA pairs related to skeletal muscle were also obtained such as novel_circ_0002153-miR-12219-5p, novel_circ_0003578-miR-3064-3p and novel_circ_0010661-miR-12260-3p. This study would help to improve the regulation mechanism of circRNAs in skeletal muscle and also provide reference for broiler breeding.

**Abstract:**

Skeletal muscle growth has always been the focus of the broiler industry, and circRNAs play a significant role in this process. We collected leg muscles of slow- and fast-growing Bian chicken embryos in the study at 14 (S14 and F14) and 20 (S20 and F20) days for RNA-seq. Finally, 123 and 121 differentially expressed circRNAs (DECs) were identified in S14 vs. F14 and S20 vs. F20, respectively. GO enrichment analysis for DECs obtained important biological process (BP) terms including nicotinate nucleotide biosynthetic process, nicotinate nucleotide salvage, and NAD salvage in S20 vs. F20 and protein mannosylation in S14 vs. F14. KEGG pathway analysis showed Wnt signaling pathway, Tight junction, Ubiquitin mediated proteolysis, and Notch signaling pathway were enriched in the top 20. Based on the GO and KEGG analysis results, we found some significant host genes and circRNAs such as NAPRT and novel_circ_0004547, DVL1 and novel_circ_0003578, JAK2 and novel_circ_0010289, DERA and novel_circ_0003082, etc. Further analysis found 19 co-differentially expressed circRNAs between the two comparison groups. We next constructed a circRNA-miRNA network for them, and some candidate circRNA-miRNA pairs related to skeletal muscle were obtained, such as novel_circ_0002153-miR-12219-5p, novel_circ_0003578-miR-3064-3p, and novel_circ_0010661-miR-12260-3p. These results would help to reveal the mechanism for circRNAs in skeletal muscle and also provide some guidance for the breeding of broilers.

## 1. Introduction

China is a major country for chicken consumption. With the improvement in people’s living standards, the demand for high-quality yellow feather broilers is becoming increasingly urgent. However, the growth rates of local chicken breeds in China are slow, and the feed utilization rate is low, so molecular breeding is needed to accelerate the genetic improvement of growth traits. Research on the genetic basis and regulation mechanism of growth traits of chickens will become an essential theoretical basis and premise of molecular breeding. The numbers of myofibers in birds are fixed late in embryonic development. Primary myofibers in chickens are formed around the embryonic age of 6 days, and secondary myofibers begin to differentiate at the embryonic age of 12–16 days, accounting for a majority of myofibers, which are basically fixed after the 20-day embryonic age [1,2]. Hence, the embryonic stage is an important period for the skeletal muscle development of birds.

Circular RNAs (circRNAs) are biologically active nucleic acid molecules that are produced by precursor mRNA back-splicing in organisms [3,4]. The first RNA molecule with a circular structure in a potato affected by spindle tuber disease was discovered in 1971 [5]. Five years later, Sanger et al. [6] found that some viroids in plants are single-stranded, covalently closed circular RNA molecules. However, circRNAs have not received much attention until Salzman et al. [7] proved that a substantial fraction of the spliced transcripts from hundreds of genes were circular RNAs by deep sequencing of RNA in 2012, and Memczak et al. [8] further revealed that circRNA CDR1as functions to bind miR-7 in neuronal tissues in 2013. Then, rapid advances in the next generation sequencing and related bioinformatics methods accelerated research on circRNAs, and thousands of circRNAs have been isolated and identified in species [7,8,9,10].

In recent years, circRNAs have been found to be involved in regulating many biological functions [11], including skeletal muscle growth [12]. Extensive research showed that circRNAs mainly served as miRNA sponges to affect biological function. Zhu et al. [13] observed that circRNA FUT10 was highly expressed in aged skeletal muscle stem cells of mice and suppressed cell proliferation and differentiation by competitively binding miR-365a-3p to attenuate the suppressive effect of miR-365a-3p on HOXA9. Das et al. [14] revealed that circNfix promoted MEF2C expression during muscle cell differentiation in part by acting as a sponge for miR-204-5p in the C2C12 mouse myoblast cell line. In addition, some circRNAs can exert their functions by translating proteins. Legnini et al. [15] identified the circRNA circ-ZNF609 with an open reading frame spanning that can be translated into a protein, resulting in specifically controlling murine and human myoblast proliferationIdentification and research on circRNAs provided a new perspective for the study of chicken skeletal muscle growth and development. Bian chicken is an eminent native Chinese breed that has good meat quality and strong adaptability. However, the growth rate of this chicken breed is slower than that of other local chicken breeds in China. In the experiment, we collected leg muscles of slow- and fast-growing Bian chicken embryos at the ages of 14 and 20 days for RNA-seq. Some key circRNAs were expected to be revealed for chicken muscle development. This study will help to improve the regulation mechanism of circRNAs in skeletal muscle and also provide a reference for broiler breeding.

## 2. Materials and Methods

### 2.1. Animals and Tissues

The slow- and fast-growing groups of Bian chicken were established by our team with myostatin gene-assisted selection [16]. Then, the bidirectional selection of body weight at 16 weeks was further carried out for six generations. The 16-week body weights of female fast-growing and slow-growing Bian chickens in the seventh generation were 1615 ± 176 g and 921 ± 93 g, respectively. At the age of 300 days, twelve female and one male Bian chicken in the seventh generation with average weight were selected. Fertilized eggs were collected after artificial insemination from the two groups and then incubated to 14 and 20-day embryo ages (14E and 20E). We dissected the chicken embryos and preliminarily determined gender by gonadal observation because the right gonad of the female chicken would degenerate. Then, the CHD1 gene was amplified using allantoic fluid (14E) or the full blood (20E) to further identify the sex of the chicken embryos at the molecular level. Agarose gel electrophoresis of PCR products showed two bands for female chickens and one band for male chickens (Appendix A). Finally, the left leg muscles of female Bian chicken embryos were collected and frozen in liquid nitrogen immediately for RNA extraction later. We designed four experimental groups with four biological replicates: 14-day embryos of the slow- and fast-growing groups (S14/F14) and 20-day embryos of the slow- and fast-growing groups (S20/F20).

### 2.2. Construction of cDNA Library

Total RNAs were extracted from leg muscles. The sequencing cDNA libraries were then generated with the qualified RNAs in which ribosomal RNA (rRNA) was first removed [17]. To select cDNA fragments of preferentially 250~300 bp in length, the library fragments were purified with the AMPure XP system. Uridine digestion was performed using Uracil-N-Glycosylase, which was followed by cDNA amplification using PCR. Finally, the qualified cDNA libraries were sequenced on the Illumina platform (NovaSeq 6000), and 150 bp paired-end reads were generated.

### 2.3. Bioinformatics Analysis for the Sequencing Data

Raw data (raw reads) of FASTQ format were processed, and clean data (clean reads) were obtained by removing reads containing adapter and ploy-N and removing low-quality reads from raw data. At the same time, Q20, Q30, and GC content of clean data were calculated. All of the downstream analyses were based on the clean data with high quality.

Paired-end clean reads were aligned to the reference genome GRCg6a using Bowtie 2 (v2.2.8) [18]. The circRNAs were then detected and identified using find_circ [8] and CIRI2 (v2.0.5) [19]. Differential expression analysis between fast- and slow-growing groups was performed using DESeq2 [20]. CircRNAs with *p*-value ≤ 0.05 were assigned as differentially expressed circRNAs (DECs) in the study. Finally, the gene ontology (GO) enrichment and KEGG pathway analysis of the host genes for DECs were implemented with GOseq (R-3.3.2) and KOBAS (v2.0), respectively.

### 2.4. Construction of circRNA–miRNA Interaction Network

It is an important mechanism for circRNAs to function by regulating downstream target genes through recruiting miRNAs. MicroRNA target sites in exons of the 19 co-differentially expressed circRNAs were identified using miRanda software. GO enrichment analysis of miRNAs was then directly performed on the Novomagic platform (https://magic.novogene.com) (accessed on 13 September 2022). Finally, genes of GO_BP terms enriched in skeletal muscle development were selected for interaction network with Cytoscape (3.8.2).

### 2.5. Validation for Differentially Expressed circRNAs

Five differentially expressed circRNAs (DECs) were selected for validation of RNA-seq in the two comparison groups (F14 vs. S14, F20 vs. S20). Specific divergent primers were designed with Primer 5.0 to confirm back-spliced junction (BSJ) sites of DECs. The PCR products were detected by agarose gel electrophoresis and sequenced with Sanger sequencing in Sango Biotech Co. Ltd. (Shanghai, China). RT-qPCR was performed on the Applied Biosystems 7500 platform (Applied Biosystems, USA) with reagent Taq Pro Universal SYBR qPCR Master Mix (Q712, Vazyme Biotech Co., Ltd., Nanjing, China). Gene β-actin was used as the housekeeping gene, and relative expression was calculated with the 2^−ΔΔCT^ method.

### 2.6. Statistical Analysis

Significant differences in body weight between slow- and fast-growing chickens were calculated using SPSS13.0 software with the Independent Samples *t* Test method. All data were presented as mean ± standard error (SE).

## 3. Results

### 3.1. Statistics of Chicken Egg Weight and Embryo Bodyweight

Average egg weights before incubation were recorded as follows: S14: 46.07 ± 0.50 (g), F14: 54.50 ± 0.38 (g), S20: 45.89 ± 0.84 (g), F20: 56.16 ± 1.16 (g). Egg weights showed a significant difference in both comparison groups S14 vs. F14 (*p* = 0.000) and S20 vs. F20 (*p* = 0.000). The bodyweights of slow- and fast-growing embryos at 14 and 20 days were as follows: S14: 9.26 ± 0.32 (g), F14: 11.25 ± 0.32 (g), S20: 32.60 ± 0.85 (g) and F20: 39.26 ± 1.20 (g). A significant difference in bodyweight was found between the slow- and fast-growing groups of 14-day embryos (S14 vs. F14; *p* = 0.005), as well as between 20-day embryos (S20 vs. F20; *p* = 0.004).

### 3.2. Summary of Sequencing Data

There were 204.55 Gb of raw data in total, and 202.93 Gb of clean data were obtained after quality control (Appendix A). The percentage of the clean base with Q20 and Q30 was at least 98.00% (F20_4) and 93.99% (F20_4), respectively. The GC content of all of the samples was about 46.60%.

Total clean reads mapped to the genome (Appendix A) were more than 91.02% (F20_2). Data alignment regions covered exons, introns, and intergenic regions, of which exon regions accounted for the highest proportion of all of the mapped clean reads.

### 3.3. Statistics of Differentially Expressed circRNAs

A total of 225 differentially expressed circRNAs (DECs) were identified with *p*-value ≤ 0.05, including 123 DECs (Appendix A) in comparison group S14 vs. F14, 121 DECs (Appendix A) in comparison group S20 vs. F20, and 19 co-differentially expressed circRNAs (Figure 1). Compared with S14, there were 51 upregulated and 72 downregulated DECs in group F14 (Figure 1a). Similarly, 51 upregulated and 70 downregulated DECs were found in group F20 compared with S20 (Figure 1b). Cluster analysis of DECs was then performed according to the expression, and results (Figure 2) showed that samples in the same group were well clustered together (S14 vs. F14 and S20 vs. F20).

### 3.4. Functional Analysis of the Host Genes of Differentially Expressed circRNAs

GO function enrichment analysis found 708 (Appendix A) and 202 (Appendix A) significantly enriched GO terms (*p*-value ≤ 0.05) in S14 vs. F14 and S20 vs. F20, respectively. Biological process (BP) terms in the top 30 of S14 vs. F14 contained formation of the primary germ layer, horizontal cell localization, positive regulation of transcription involved in the G1/S phase of the mitotic cell cycle, and regulation of cellular catabolic process (Figure 3a). In the comparison group, S20 vs. F20, BP terms such as nicotinate nucleotide biosynthetic process, nicotinate nucleotide metabolic process, protein maturation, and cellular protein metabolic process were found in the top 30 (Figure 3b).

KEGG pathway analysis (Appendix A) showed that six of the top 20 KEGG pathways in the two comparison groups were the same, including Fanconi anemia pathway, regulation of actin cytoskeleton, pentose phosphate pathway, tight junction, herpes simplex infection and ubiquitin-mediated proteolysis (Figure 4). In addition, some other vital pathways were also found in the top 20 pathways, such as the mTOR signaling pathway, Apoptosis, Wnt signaling pathway, MAPK signaling pathway and Focal adhesion in group S14 vs. F14, and Protein export, Nicotinate and nicotinamide metabolism, Jak-STAT signaling pathway, Notch signaling pathway and Metabolic pathways in group S20 vs. F20.

### 3.5. CircRNA–miRNA Interaction Network

As their main mechanism, circRNAs may act as miRNA sponges to modulate post-transcriptional regulation. A total of 55 GO_BP terms related to skeletal muscle development were found (Appendix A) for miRNAs of the 19 co-differentially expressed circRNAs. The miRNA–circRNA interaction network was then constructed with Cytoscape (3.8.2), and circRNA with a high “degree” played an important role. Figure 5 (Table 1) shows that the degree values of novel_circ_0002153 and novel_circ_0003578 were the highest, followed by novel_circ_0010661, novel_circ_0003082, and novel_circ_0005681. These circRNAs may regulate the growth of skeletal muscle through the ceRNA mechanism, and they will also be the focus of functional experiments in future cell experiments.

### 3.6. Verification of circRNA Sequencing Results

PCR product sizes of divergent primers for the five DECs were consistent with that designed by Primer 5.0 after agarose gel electrophoresis (Figure 6a). Sanger sequencing results for PCR products showed that the bases were the same as those of RNA-seq (Figure 6b). Finally, the relative expression trend of RT-qPCR for the five DECs between the fast- and slow-growing groups also corresponded to that of RNA-seq (Figure 6c). The above experiments indicated that the results with RNA-seq are reliable.

## 4. Discussion

Animal meats provide a large amount of protein for humans worldwide, among which chicken plays an important role [21]. It has the characteristics of high protein, low fat, and low calories. Chicken is also more cost-effective compared to other meat sources [22]. Skeletal muscle growth in the broiler industry is particularly significant because it can directly affect chicken yield [23,24]. CircRNAs have been extensively studied in recent years, and their function in chicken skeletal muscle growth has also been gradually revealed. Li et al. [25] observed that overexpression of circTAF8 could promote the proliferation of chicken primary myoblasts and inhibit their differentiation. In addition, a total of eight SNPs in the introns flanking the circTAF8 locus were associated with carcass traits such as leg muscle weight, live weight, and half- and full-bore weight. Zhao et al. [26] revealed that the newly identified circCCDC91 promoted myoblast proliferation and differentiation and alleviated skeletal muscle atrophy by directly binding to the miR-15 family via activating the IGF1-PI3K/AKT signaling pathway in chicken. Yin et al. [27] found a novel circular RNA circFAM188B that encodes a novel protein circFAM188B-103aa to promote proliferation and inhibit differentiation in chicken skeletal muscle satellite cells.

The embryonic stage of chicken is a significant period for the initial formation of muscle fibers [28,29]. CircRNA studies provide a new perspective for the exploration of embryonic skeletal muscle development. In the experiment, we collected leg muscles of fast- and slow-growing Bian chickens at 14- and 20-day embryo ages for RNA-seq. Finally, a total of 123 and 121 DECs were found in S14 vs. F14 and S20 vs. F20, respectively. Our previous research showed that the c.234G > A mutation of the myostatin gene had a significant effect on the growth traits of female Bian chickens. Based on this mutation, fast-growing (AA genotype) and slow-growing (GG genotype) strains were successfully established [16]. Then, the bidirectional selection of body weight at 16 weeks was further carried out for six generations. We surveyed of all the DECs detected in the study and found that no DEC was produced by the myostatin gene. We speculate that the myostatin gene exerts its effect on the growth traits of Bian chicken through mRNA and protein rather than circRNA. GO enrichment and KEGG pathway analysis were then performed for the host genes of DECs.

Some GO_BP terms related to nicotinamide adenine dinucleotide (NAD) in comparison S20 vs. F20 were found, including nicotinate nucleotide biosynthetic process, nicotinate nucleotide salvage, pyridine nucleotide salvage, NAD salvage, and nicotinate nucleotide metabolic process. NAD serves both as a coenzyme for redox reactions, making it central to energy metabolism, and as an essential cofactor for non-redox NAD-dependent enzymes such as sirtuins. It can influence many cellular functions, including metabolic pathways, DNA repair, chromatin remodeling, cellular senescence, and immune cell function [30], and skeletal muscle growth is also closely related to NAD [31,32]. NAD can be synthesized from the different dietary precursors by the de novo pathway (tryptophan), the Preiss–Handler pathway (nicotinic acid, NA), and the salvage pathway (nicotinamide riboside, NR or nicotinamide) [33]. Nicotinamide mononucleotide (NMN) is generated by the NAD salvage pathway, while nicotinic acid mononucleotide (NAMN) and nicotinic acid adenine dinucleotide (NAAD) are generated by both the de novo and Preiss–Handler NAD pathways [34]. NAMN is structurally very similar to NMN, differing only at the C3 position of the pyridine, where the amide group in NMN is replaced by the carboxyl group in NAMN [35]. The major dietary source of NAD is nicotinic acid, and it could be transformed into NAD through three steps in the Preiss–Handler pathway [34], in which nicotinic acid phosphoribosyltransferase (NAPRT) is a necessary enzyme that can convert NA into NAMN in the first step [36]. It was enriched in several NAD-related BP terms in the study, and the corresponding circRNA is novel_circ_0004547, which may be a key candidate circRNA for skeletal muscle development.

In addition, some metabolic items are also greatly affected by NAD, including the primary metabolic process and cellular macromolecule metabolic process in S20 vs. F20 and regulation of the cellular catabolic process in S14 vs. F14. The genes enriched in them and their corresponding circRNAs may be very important for skeletal muscle, such as DVL1 and JAK2. DVL1, the host gene of novel_circ_0003578, is enriched in the above three items and has been found to inhibit human myoblasts [37]. The host gene JAK2 of novel_circ_0010289 was found in both BP terms of S20 vs. F20, and it has also been revealed to regulate skeletal muscle development [38,39].

Another category of BP terms that accounts for a large proportion in the top 30 are “protein” related entries, including protein mannosylation in S14 vs. F14, and protein folding in the endoplasmic reticulum, protein maturation, chaperone-mediated protein folding, negative regulation of protein activation cascade, cellular protein metabolic process and protein folding in S20 vs. F20. Skeletal muscle is one of the most dynamic and plastic tissues. It comprises approximately 40% of total body weight and contains 50–75% of all body proteins [40]. Skeletal muscle protein plays important roles in many functions, including skeletal muscle atrophy, impaired muscle growth or regrowth, and functional decline [41]. ERO1A was enriched in all protein BP terms except negative regulation of protein activation cascade in S20 vs. F20, followed by genes SUCO and FKBP51. The DECs produced from them are novel_circ_0008046, novel_circ_0009748, and novel_circ_0003926, respectively. In addition, TMTC2 was enriched in BP terms in both of the two comparison groups, and its corresponding circRNAs are novel_circ_0002879 and novel_circ_0002881 in S14 vs. F14 and novel_circ_0002882 in S20 vs. F20. The above circRNAs may potentially regulate the development of skeletal muscle.

KEGG enrichment analysis showed that most of the pathways in the top 20 of the two comparison groups were related to skeletal muscle development, among which the same items included regulation of actin cytoskeleton, tight junction, ubiquitin-mediated proteolysis, and pentose phosphate pathway. Some KEGG pathways for host genes of DECs in the experiment were the same as those of other studies. Pan et al. [42] collected the leg muscle from male embryos of Tibetan chicken at embryonic (E) days 10 and 18 for RNA sequencing, and KEGG results for the host genes of DECs revealed that the enriched pathways involved Wnt signaling pathway and MAPK signaling pathway. The two terms were also enriched in the top 20 KEGG pathways in S14 vs. F14. In another study of circRNAs for the longissimus dorsi of different cattle breeds, Liu et al. [9] enriched some KEGG pathways related to muscle development, such as the MAPK signaling pathway, Wnt signaling pathway, and Regulation of actin cytoskeleton and mTOR signaling pathway, which were also enriched in our research. We focused on the host genes of DECs with the highest enrichment frequency in skeletal muscle growth-related pathways in the top 20.

Protein kinase C alpha (PRKCA), a subtype of protein kinase C (PKC), was enriched in 8 items in S14 vs. F14, including the mTOR signaling pathway, Wnt signaling pathway, and MAPK signaling pathway. PKCs are involved in multiple signal transduction systems that control cell proliferation, differentiation, survival, invasion, migration, and apoptosis [43]. Studies have shown that PRKCA (PKCα) influences muscle fibers’ formation and skeletal muscle development [44,45]. Therefore, its corresponding circRNA novel_circ_0001545 may potentially affect skeletal muscle growth.

PPP3CB, the host gene of novel_circ_0008243, was enriched in Apoptosis, Wnt signaling pathway, MAPK signaling pathway, and Calcium signaling pathway in S14 vs. F14. It belongs to the protein phosphatase family, which could catalyze the dephosphorylation proteins. Protein phosphatases and kinases control multiple cellular events, including proliferation and differentiation, through regulating reversible protein phosphorylation, the most important post-translational modification [46]. PPP1CB is also a member of protein phosphatases and was enriched in the Regulation of actin cytoskeleton and Focal adhesion pathways of S14 vs. F14. It is the host gene of novel_circ_0005922, which should also be valued in skeletal muscle.

In comparison group S14 vs. F14, PTEN was enriched in the mTOR signaling pathway, Phosphatidylinositol signaling system, Tight junction and Focal adhesion. PTEN is a dual phosphatase with both protein and lipid phosphatase activities [47]. It is a major negative regulator of the signaling pathway defined by class I phosphoinositide 3 kinase (PI3K), AKT, and the mechanistic target of rapamycin (mTOR) and controls a wide range of essential cellular processes, including cell proliferation, growth, survival, and metabolism [48]. Studies have revealed that PTEN inhibition could prevent skeletal muscle atrophy and improve muscle integrity and function [49,50]. The circRNA (novel_circ_0008144) also potentially regulates skeletal muscle development.

Three genes, DERA, ARHGEF7, and MGRN1, were enriched in the Pentose phosphate pathway, Regulation of actin cytoskeleton and Ubiquitin mediated proteolysis pathways in both of the comparison groups. Park et al. [51] identified DERA as a core gene in the postmortem energy metabolism of longissimus dorsi muscle (LDM) in the pig. The PIX proteins, ARHGEF6 (α-PIX) and ARHGEF7 (β-PIX), are guanine nucleotide exchange factors (activators) for the Rho family small GTP-binding protein family members, and they could control cell polarity, adhesion, and migration in the pathway [52]. MGRN1 acts as an E3 ubiquitin ligase [53], and the ubiquitin-proteasome system could affect muscle mass, including myofibrillar protein degradation and myogenesis inhibition [54]. The three genes and their DECs (novel_circ_0003082 and novel_circ_0003084, novel_circ_0002195 and novel_circ_0002200, novel_circ_0000876) may also affect the growth and development of skeletal muscle in chicken.

DVL1 and NAPRT have been discussed in the above BP item part. KEGG enrichment analysis showed that DVL1 was enriched in the Wnt signaling pathway of S14 vs. F14 and Notch signaling pathway of S20 vs. F20, while NAPRT was found in the Nicotinate and nicotinamide metabolism and Metabolic pathways in S20 vs. F20.

CircRNAs can recruit miRNAs to regulate target gene expression [55], which is one of the main regulatory mechanisms for circRNAs. Shen et al. [56] identified an abundant circular RNA circTMTC1 by RNA-seq and further proved that it could inhibit chicken skeletal muscle satellite cell differentiation by sponging miR-128-3p. Liu et al. [57] found that circARID1A could regulate skeletal muscle cell development and regeneration by sponging miR-6368. In this study, we performed GO enrichment analysis for all predicted miRNAs of the 19 co-differentially expressed circRNAs and constructed a circRNA-miRNA network with the skeletal muscle-related BP terms. Five key circRNAs (novel_circ_0002153, novel_circ_0003578, novel_circ_0010661, novel_circ_0003082, and novel_circ_0005681) were screened according to the degree. Each circRNA was linked to multiple miRNAs, and it may regulate the growth of skeletal muscle through circRNA-miRNA pairs such as novel_circ_0002153-miR-12219-5p, novel_circ_0003578-miR-3064-3p, novel_circ_0010661-miR-12260-3p, novel_circ_0003082-miR-12282-3p, and novel_circ_0005681-miR-1687-3p.

## 5. Conclusions

In this experiment, we identified 123 and 121 DECs in comparison groups S14 vs. F14 and S20 vs. F20, respectively. Based on the important GO terms and KEGG pathways, we found some significant circRNAs. In addition, the circRNA-miRNA network of the 19 co-differentially expressed circRNAs revealed some skeletal muscle-related circRNA-miRNA pairs. These findings will guide our next research work and further help to reveal the development mechanism of circRNA in chicken skeletal muscle.

## Figures and Tables

**Figure 1 animals-12-03166-f001:**
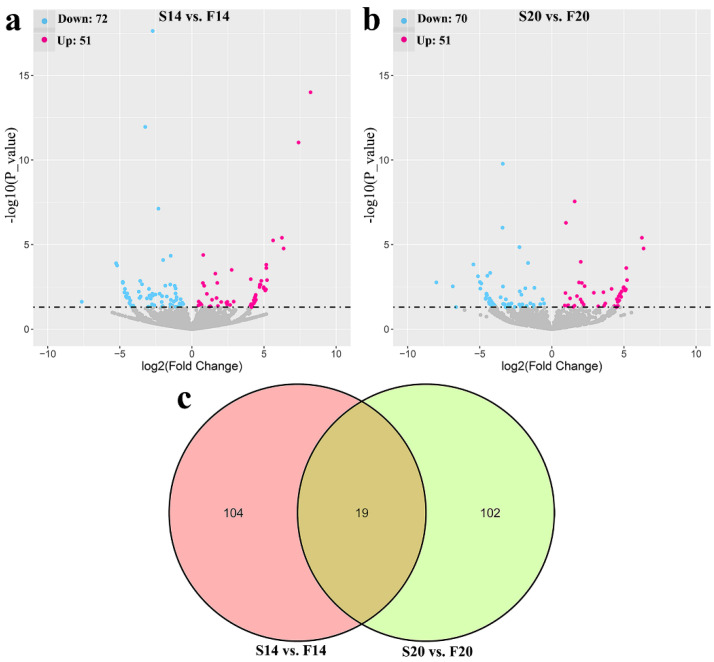
Statistics of differentially expressed circRNAs (DECs). (**a**) Volcano plot of DECs in S14 vs. F14 group. (**b**) Volcano plot of DECs in S20 vs. F20 group. (**c**) Venn diagram of DECs.

**Figure 2 animals-12-03166-f002:**
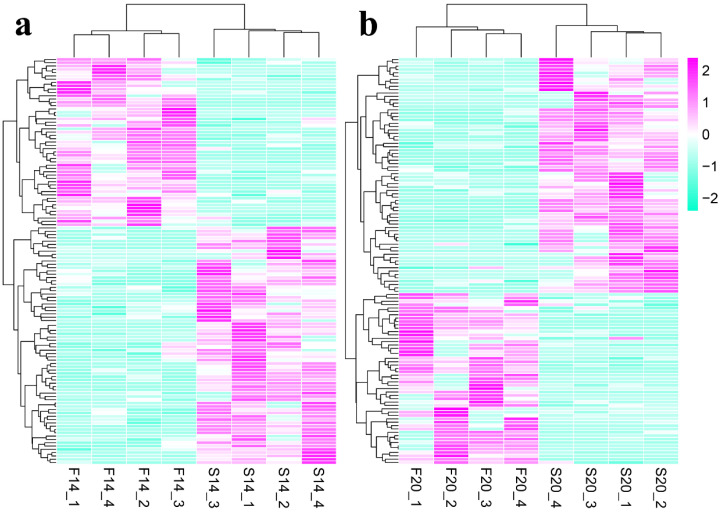
Clustering analysis of differentially expressed circRNAs (DECs). (**a**) Heatmap for DECs of S14 vs. F14 group. (**b**) Heatmap for DECs of S20 vs. F20 group.

**Figure 3 animals-12-03166-f003:**
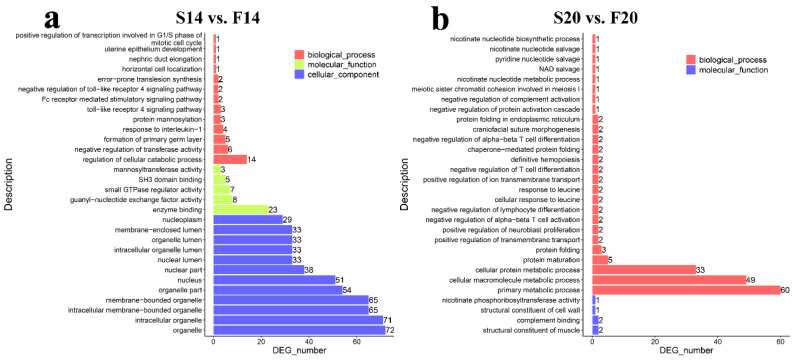
GO enrichment analysis of the host genes of the differentially expressed circRNAs. (**a**) Results of GO analysis for S14 vs. F14. (**b**) Results of GO analysis for S20 vs. F20.

**Figure 4 animals-12-03166-f004:**
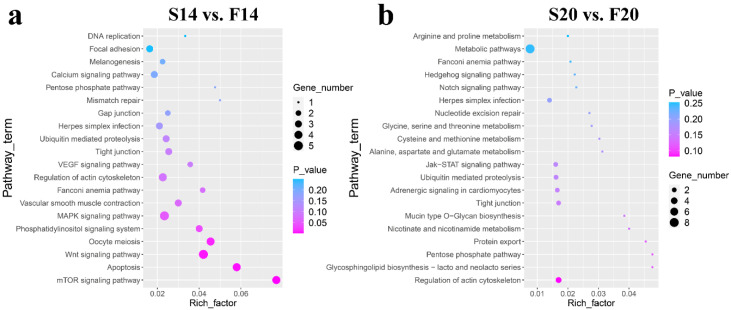
KEGG pathway analysis of the host genes of the differentially expressed circRNAs. (**a**) Results of KEGG analysis for S14 vs. F14. (**b**) Results of KEGG analysis for S20 vs. F20.

**Figure 5 animals-12-03166-f005:**
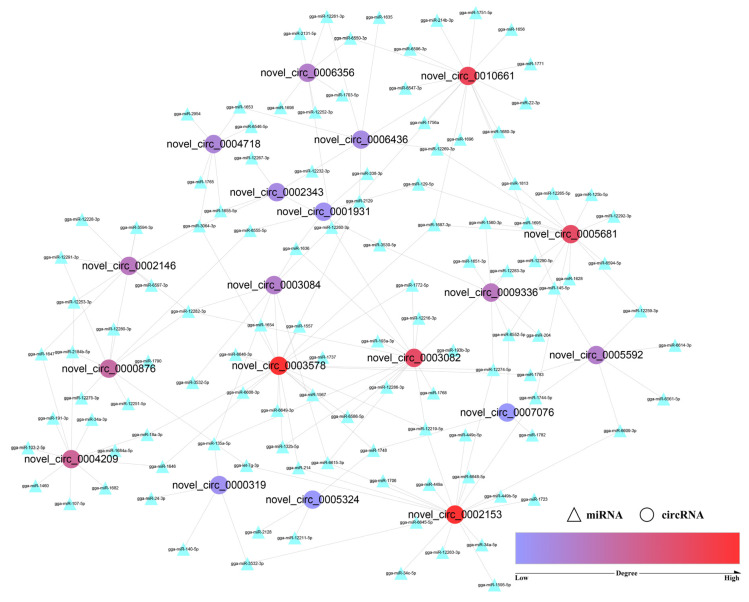
circRNA–miRNA interaction network. Note: Circular nodes represent circRNAs, and the triangle nodes represent miRNAs. The degrees of red nodes are high, while the degrees of blue nodes are low.

**Figure 6 animals-12-03166-f006:**
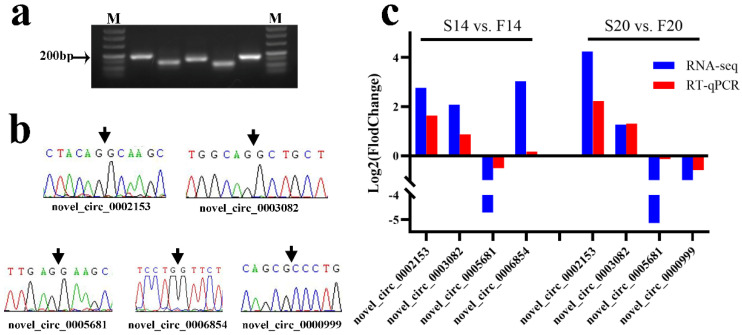
Validation of differentially expressed circRNAs with RNA-seq. (**a**) Agarose gel electrophoresis results for PCR products of divergent primers with cDNA; M: Maker. The electrophoretic bands from left to right are novel_circ_0002153, novel_circ_0003082, novel_circ_0005681, novel_circ_0006854, and novel_circ_0000999. (**b**) Sanger sequencing results for the back-splicing junction of circRNAs. (**c**) RT-qPCR validation of five differentially expressed circRNAs in S14 vs. F14 and S20 vs. F20, respectively. NOTE: Original gel for Figure 6a is presented in Appendix A.

**Table 1 animals-12-03166-t001:** 19 Co-differentially expressed circRNAs with their degrees.

Name	Host Gene	Degree	Name	Host Gene	Degree
novel_circ_0002153	AHCYL2	19	novel_circ_0005592	ITSN2	7
novel_circ_0003578	DVL1	19	novel_circ_0006356	REV3L	7
novel_circ_0010661	ENSGALG00000002326	16	novel_circ_0004718	SLC25A13	6
novel_circ_0003082	DERA	15	novel_circ_0002343	DIS3	5
novel_circ_0005681	CDC5L	15	novel_circ_0006436	RRAGD	5
novel_circ_0004209	NCOA2	11	novel_circ_0000319	GPATCH1	4
novel_circ_0000876	MGRN1	10	novel_circ_0001931	CNKSR2	4
novel_circ_0002146	ENSGALG00000016826	8	novel_circ_0005324	MOCOS	3
novel_circ_0009336	LRRFIP1	8	novel_circ_0007076	MOB1B	3
novel_circ_0003084	DERA	7			

## Data Availability

The raw data of the study have been uploaded to the Sequence Read Archive (SRA) database, and the accession number is PRJNA773377 (https://www.ncbi.nlm.nih.gov/bioproject/?term=PRJNA773377) (accessed on 21 October 2021).

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
