# Peer review of "Transcriptome Sequencing Analysis of circRNA in Skeletal Muscle between Fast- and Slow-Growing Chickens at Embryonic Stages"

_animals, 2022, doi:10.3390/ani12223166_

Round 1
Reviewer 1 Report
In this manuscript the authors have nicely showed a number of differentially expressed circRNAs between fast and slow growing chickens during skeletal muscle embryonic development. Results suggested association of these circRNAs with some important Biological Process terms and with KEGG pathways associated with muscle growth and maintenance. They have also showed the possible circRNA-miRNA interaction network, which suggested the association of novel circRNAs with muscle growth.
The work was well performed and shows nice results. For this reason, this reviewer suggests it to be accepted to be published in Animals journal after few minor revisions:
1. Lines 65 – 66: The authors described that “the left leg muscles of female Bian chicken embryos” were collect for RNA isolation. How was the sex determined in the embryos?
2. In which moment was the embryos weighted? Were they weighted free of extraembryonic vesicles?
3. Lines 147 – 148: “cellular protein metabolic process” is repeated.
4. Figure 6a: PCR products seem to be not specific, especially for the first three primers. Was RT-qPCR based on these primers?
5. “Conclusions” section contains a list of the main results. Please, review the text showing only the main findings of the data.
Author Response
- Lines 65 – 66: The authors described that “the left leg muscles of female Bian chicken embryos” were collect for RNA isolation. How was the sex determined in the embryos?
Answer:We dissected the chicken embryos and preliminarily determined gender by gonadal observation because the right gonad of the female chicken would degenerate. Then the CHD1 gene was amplified using allantoic fluid (14E) or the full blood (20E) to further identify the sex of chicken embryos at the molecular level. Agarose gel electrophoresis of PCR products showed two bands for female chickens and one band for male chickens (Figure S1). (lines 79-84)
- In which moment was the embryos weighted? Were they weighted free of extraembryonic vesicles?
Answer:The bodyweights of slow- and fast-growing embryos at 14 and 20 days were recorded. I have added the time for weights. (lines 86-88; line 135). And embryos were weighted free of extraembryonic vesicles.
- Lines 147 – 148: “cellular protein metabolic process” is repeated.
Answer:Thank you. I have modified it. (Line 170)
- Figure 6a: PCR products seem to be not specific, especially for the first three primers. Was RT-qPCR based on these primers?
Answer:Yes. RT-qPCR was also based on these primers and the melt curve for each primer have a unique peak. We conducted agarose gel electrophoresis again, and uploaded the picture (Figure 6).
- “Conclusions” section contains a list of the main results. Please, review the text showing only the main findings of the data.
Answer:I have revised the conclusion and showed the main findings of the data. (lines 355-360)
Reviewer 2 Report
The manuscript titled “Transcriptome sequencing analysis of circRNA in skeletal muscle between fast- and slow-growing chickens at embryonic stages” compared the expression profile of the circular RNA between fast- and slow-growing leg muscle at embryonic 14th day and embryonic 20th day, identified the DECs and the co-differentially expressed circRNAs between the S14 vs. F14 and S20 vs. F20, constructed circRNA-miRNA network with the co-differentially expressed circRNAs. The materials used in the research are sufficient. This study provides some information for circRNAs expression in embryonic skeletal muscle growth and development. However, some results are needed to be presented in the manuscript and some key information are needed.
General comments:
1. In the RNA-seq project, the differentially expressed genes were usually identified using p < 0.05 and |log2FC| > 1 as cutoffs. In current study, circRNAs with P-value ≤ 0.05 were assigned as DECs. Is it appropriate? Since there are no supplementary tables in the system, the number of DECs with p < 0.05 and |log2FC| > 1 can’t be determined.
2. In the Material and Method part, some key information should be added. For example, the information on the sex of the embryos. Did the authors use the embryo with the same sex for RNA-seq? If so, provide the method for the sex determination. The authors provided the information of the average body weight in the Results part, I think the egg weight before the incubation should be provided.
3. The authors presented the circRNA-miRNA network with the 19 co-differentially expressed circRNAs, I think the other DECs should be also considered and presented. Because the main goal of this project is to investigate the DECs between fast- and slow-growing chickens. The myoblasts are differentiating at the 14th embryonic day of chicken. On the 20th embryonic day, the skeletal muscle has completed the differentiation and the hypertrophy of the muscle fiber is the main reason of body weight gain. The 14th and 20th embryonic day are the two distinguished stages of skeletal muscle development in chickens. So, the specific DECs in each comparison should be preset in the Results and Discussion part.
Minor comments:
1. Line 9, delete “Correspondence”.
2. L16, “S20vsF20”, spaces should be added before and after the “vs”, change “vs” to “ vs.”. Check the whole context, Figures and Tables.
3. L85, change the “The circRNA were” to “The circRNAs were”.
4. L119, replace “G” with “Gb”.
5. The Results part 3.2, the detail information of quality control and genome alignment should be provided.
6. Can’t find Table S1-S7.
7. Figure 5, Please provide a better figure with high resolution.
8. Figure 6b, the peaks of sequencing results should be provided integrally.
9. Table 1, provide the parental genes of the co-differentially expressed circRNAs.
10. The structure and format of the manuscript should be checked thoroughly.
Author Response
General comments:
- In the RNA-seq project, the differentially expressed genes were usually identified using p < 0.05 and |log2FC| > 1 as cutoffs. In current study, circRNAs with P-value ≤ 0.05 were assigned as DECs. Is it appropriate? Since there are no supplementary tables in the system, the number of DECs with p < 0.05 and |log2FC| > 1 can’t be determined.
Answer:I have provided all the supplementary tables at present. The DECs were shown in Table S3 and S4. The screening standard with DESeq2 (p < 0.05 and |log2FC| > 1) for DECs is an experience value and they can be adjusted up and down appropriately. Log2FC ≥ 0 was also used with software DESeq2 by Monroe et al. [1] in RNA-seq. Zhang et al. [2] applied DESeq2 to identify coding-genes and long intergenic noncoding RNAs (lincRNAs) differentially expressed in MI and high CAC without limiting log2FC. In addition, we found that DECs with |log2FC| > 1 in the study were more than 90% (Table S3 and S4). Therefore, DECs found in the study are credible.
[1] Monroe JD, Moolani SA, Irihamye EN, Speed JS, Gibert Y, Smith ME. RNA-Seq Analysis of Cisplatin and the Monofunctional Platinum(II) Complex, Phenanthriplatin, in A549 Non-Small Cell Lung Cancer and IMR90 Lung Fibroblast Cell Lines. Cells. 2020 Dec 8;9(12):2637. doi: 10.3390/cells9122637. PMID: 33302475; PMCID: PMC7764052.
[2] Zhang X, van Rooij JGJ, Wakabayashi Y, Hwang SJ, Yang Y, Ghanbari M, Bos D; BIOS Consortium, Levy D, Johnson AD, van Meurs JBJ, Kavousi M, Zhu J, O'Donnell CJ. Genome-wide transcriptome study using deep RNA sequencing for myocardial infarction and coronary artery calcification. BMC Med Genomics. 2021 Feb 10;14(1):45. doi: 10.1186/s12920-020-00838-2. PMID: 33568140; PMCID: PMC7874462.
- In the Material and Method part, some key information should be added. For example, the information on the sex of the embryos. Did the authors use the embryo with the same sex for RNA-seq? If so, provide the method for the sex determination. The authors provided the information of the average body weight in the Results part, I think the egg weight before the incubation should be provided.
Answer:We used the embryos with the same sex (female) for RNA-seq (line 84). And I have also provided the method for the sex determination (line 79-83). The egg weight before the incubation were also provided. (lines 132-134).
- The authors presented the circRNA-miRNA network with the 19 co-differentially expressed circRNAs, I think the other DECs should be also considered and presented. Because the main goal of this project is to investigate the DECs between fast- and slow-growing chickens. The myoblasts are differentiating at the 14th embryonic day of chicken. On the 20th embryonic day, the skeletal muscle has completed the differentiation and the hypertrophy of the muscle fiber is the main reason of body weight gain. The 14th and 20th embryonic day are the two distinguished stages of skeletal muscle development in chickens. So, the specific DECs in each comparison should be preset in the Results and Discussion part.
Answer:Thank you for your advice. Figure 1and 2 were both performed with the DECs in each comparison. We further performed GO enrichment and KEGG pathway analysis for the host genes of DECs in each comparison (Figure 3 and Figure 4). In addition, we also discussed the specific DECs and co-DECs in S14vsF14 and S20vsF20 in the discussion section according to the results of GO and KEGG (lines 235-339).
Because there are too many DECs, we want to find more important DECs in both stage for further study. We selected the 19 co-differentially expressed circRNAs to construct circRNA-miRNA network. It does not mean that other DECs are unimportant, but these co-DECs are more widely expressed differently in the two stages. Besides, because of the large number of DECs in each stage, it is impossible to display the circRNA-miRNA network clearly for 123 DECs in comparison group S14 vs. F14, and 121 DEGs in comparison group S20 vs. F20.
Minor comments:
- Line 9, delete “Correspondence”.
Answer:Thank you. I have deleted the word. (line 9)
- L16, “S20vsF20”, spaces should be added before and after the “vs”, change “vs” to “ vs. ”. Check the whole context, Figures and Tables.
Answer:I have checked the whole context and revised them.
- L85, change the “The circRNA were” to “The circRNAs were”.
Answer:I have changed the “The circRNA were” to “The circRNAs were”. (line 103)
- L119, replace “G” with “Gb”.
Answer:I have replaced “G” with “Gb”. (line 144)
- The Results part 3.2, the detail information of quality control and genome alignment should be provided.
Answer:I have added two table S1 and S2 for the quality control and genome alignment.
- Can’t find Table S1-S7.
Answer:I have provided all Table S.
- Figure 5, Please provide a better figure with high resolution.
Answer:I have provided a better figure 5 with high resolution.
- Figure 6b, the peaks of sequencing results should be provided integrally.
Answer:Thank you for your suggestion. I have revised the Figure 6b.
- Table 1, provide the parental genes of the co-differentially expressed circRNAs.
Answer:I have provided the host gene for the co-differentially expressed circRNAs.
- The structure and format of the manuscript should be checked thoroughly.
Answer:Thank you. I have checked and revised the manuscript thoroughly.
Reviewer 3 Report
There seems no description about why the authors used chickens in the study. Therefore I could not completely understand the motivation of this study.
More and more detaild explanation of chicken samples should be required. For example, what was the Bian breed, what the terms "seventh generation" in L63 means, and why days 14 and 20 were selected for comparison?
Was using t-test valid for this study? Also, why did the authors show SD, but not SE?
What was the nuvel findings from this study? And the findings would be common to other chicken breeds and even other species? Prease discuss this point.
Author Response
There seems no description about why the authors used chickens in the study. Therefore I could not completely understand the motivation of this study.
Answer:China is a major country of chicken consumption. With the improvement of people's living standards, the demand for high-quality yellow feather broilers is becoming increasingly urgent. However, the growth rate of local chicken breeds in China is slow and the feed utilization rate is low, so molecular breeding is needed to accelerate the genetic improvement of growth traits. Research on genetic basis and regulation mechanism of growth traits of chicken will become an important theoretical basis and premise of molecular breeding. (lines 29-35).
More and more detailed explanation of chicken samples should be required. For example, what was the Bian breed, what the terms "seventh generation" in L63 means, and why days 14 and 20 were selected for comparison?
Answer: “Bian chicken is an eminent native Chinese breed, which has good meat quality, strong adaptability. However, the growth rate of this chicken breed is slow as other local chicken breeds in China. (lines 65-67).”
“The slow- and fast-growing groups of Bian chicken were established by our team with gene-assisted selection. Then the bidirectional selection of body weight was further carried out for six generations.” (lines 74-76)
“The number of myofibers in birds has been fixed late in embryonic development. Primary myofibers of chickens are formed around the embryonic age of 6, and secondary myofibers begin to differentiate at the embryonic age of 12–16, accounting for a majority of myofibers, which have been basically fixed after the 20-day embryonic age. Hence, days 14 and 20 in embryonic stage were important for the development for the skeletal muscle. (lines 35-39)”.
I have added all the information in the manuscript.
Was using t-test valid for this study? Also, why did the authors show SD, but not SE?
Answer:Yes. We used t-test valid for this study. Both SD and SE can be used to represent variation. According to your suggestion, I have changed SD to SE. (line 129; lines 132-136)
What was the nuvel findings from this study? And the findings would be common to other chicken breeds and even other species? Prease discuss this point.
Answer:We collected leg muscle of slow- and fast-growing Bian chicken embryos at embryonic age for RNA-seq. To our knowledge, this is the first study to find DECs by comparing the fast- and slow-growing embryonic skeletal muscle of in the same chicken breed. However, some KEGG pathways for host genes of DECs are the same as those of other studies. Pan et al. [1] collected the leg muscle from male embryos of Tibetan chicken at embryonic (E) 10 and E18 for RNA sequencing and KEGG results for the host genes of DECs revealed that the enriched pathways involved Wnt signaling pathway and MAPK signaling pathway. The two terms were also enriched in top20 KEGG pathways in S14 vs. F14. In another study of circRNAs for the longissimus dorsi of different cattle breeds, Liu et al. [2] enriched some KEGG pathways related to muscle development such as MAPK signaling pathway, Wnt signaling pathway, Regulation of actin cytoskeleton and mTOR signaling pathway, which are also enriched and focused in our research. (lines 288-297)
In conclusion, the study obtained new findings compared to other chicken breeds and even other species, and we also found something in common. We have also discussed this point as you suggested.
[1] Pan Z, Yang C, Zhao R, Jiang X, Yu C, Li Z. Characterization of lncRNA/circRNA-miRNA-mRNA network to reveal potential functional ceRNAs in the skeletal muscle of chicken. Front Physiol. 2022 Sep 29;13:969854. doi: 10.3389/fphys.2022.969854. PMID: 36246144; PMCID: PMC9558166.
[2] Liu R, Liu X, Bai X, Xiao C, Dong Y. Identification and Characterization of circRNA in Longissimus Dorsi of Different Breeds of Cattle. Front Genet. 2020 Nov 26;11:565085. doi: 10.3389/fgene.2020.565085. PMID: 33324445; PMCID: PMC7726199.
Round 2
Reviewer 2 Report
The manuscript can be accepted in present form.
Author Response
The manuscript can be accepted in present form.
Answer: Thank you.
Reviewer 3 Report
The authors should add the explanation about how to establish the slow- and fast-growing groups by gene-assisted selection. For example, which genes were targetted? This would be (heavily) affected the results shown in this manuscript, and the detailed and careful discussion about this point muste be required.
Author Response
The authors should add the explanation about how to establish the slow- and fast-growing groups by gene-assisted selection. For example, which genes were targetted? This would be (heavily) affected the results shown in this manuscript, and the detailed and careful discussion about this point muste be required.
Answer: The slow- and fast-growing groups of Bian chicken were established by our team with the myostatin gene assisted selection. The c.234G>A mutation of the myostatin gene (MSTN) was used in gene-assisted selection for growth traits, and it had significant effect on the growth traits of the female Bian chickens. Thus fast-growing (AA genotype) and slow-growing (GG genotype) strains were successfully established (Zhang et al., 2012; Zhang et al., 2015). Then the bidirectional selection of body weight at 16-week was further carried out for 6 generations. The body weight of female fast-growing and slow-growing Bian chickens in 7th generation was 1615±176 g and 921±93 g, respectively. (lines 76-78) At the age of 300 days, twelve female and one male Bian chickens in the seventh generation with the average weight were selected. Fertilized eggs were collected after artificial insemination from the two groups and then incubated to 14 and 20-day embryo ages. We have added the information and reference (Zhang et al., 2015) in the revised manuscript. According to your suggestion, we surveyed all the differentially expressed circRNAs. We found that no differentially expressed circRNA was produced by the myostatin gene. We speculate that the myostatin gene exerts its effect on the growth traits of Bian chicken through mRNA and protein rather than circRNA. We have discussion this point in the revised manuscript. (lines 239-247)
Zhang GX, Zhao XH, Wang JY, Ding FX, Zhang L. Effect of an exon 1 mutation in the myostatin gene on the growth traits of the Bian chicken. Animal Genetics. 2012, 43(4):458-459.
Zhang GX, Zhang T, Wei Y, Ding FX, Zhang L, Wang JY. Functional identification of an exon 1 substitution in the myostatin gene and its expression in breast and leg muscle of the Bian chicken. British Poultry Science. 2015, 56(6):639-644.